# Prehabilitation to prevent complications after cardiac surgery - A retrospective study with propensity score analysis

Johanneke Hartog[1]*, Iman Mousavi[1¤a], Sandra Dijkstra[1], Joke Fleer[2], Lucas H. V. van der Woude[3,4], Pim van der Harst[5¤b], Massimo A. Mariani[1]

1 Department of Cardiothoracic Surgery, University of Groningen, University Medical Center Groningen, Groningen, Groningen, The Netherlands, 2 Department of Health Psychology, University of Groningen, University Medical Center Groningen, Groningen, Groningen, The Netherlands, 3 Center for Human Movement Sciences, University of Groningen, University Medical Center Groningen, Groningen, Groningen, The Netherlands, 4 Department of Rehabilitation Medicine, University of Groningen, University Medical Center Groningen, Groningen, Groningen, The Netherlands, 5 Department of Cardiology, University of Groningen, University Medical Center Groningen, Groningen, Groningen, The Netherlands

¤a Current address: Department of Cardiothoracic Surgery, Onze Lieve Vrouwe Gasthuis, Amsterdam, Noord-Holland, The Netherlands
¤b Current address: Department of Cardiology, University Medical Center Utrecht, Utrecht, Utrecht, The Netherlands
* j.hartog@umcg.nl

## Abstract

### Background

The rising prevalence of modifiable lifestyle-related risk factors (e.g. overweight and physical inactivity) suggests the need for effective and safe preoperative interventions to improve outcomes after cardiac surgery. This retrospective study explored potential short-term postoperative benefits and unintended consequences of a multidisciplinary prehabilitation program regarding in-hospital complications.

### Methods

Data on patients who underwent elective cardiac surgery between January 2014 and April 2017 were analyzed retrospectively. Pearson's chi-squared tests were used to compare patients who followed prehabilitation (three times per week, at a minimum of three weeks) during the waiting period with patients who received no prehabilitation. Sensitivity analyses were performed using propensity-score matching, in which the propensity score was based on the baseline variables that affected the outcomes.

### Results

Of 1201 patients referred for elective cardiac surgery, 880 patients met the inclusion criteria, of whom 91 followed prehabilitation (53.8% ≥ 65 years, 78.0% male, median Euroscore II 1.3, IQR, 0.9–2.7) and 789 received no prehabilitation (60.7% ≥ 65 years, 69.6% male, median Euroscore II 1.6, IQR, 1.0–2.8). The incidence of atrial fibrillation (AF) was significantly lower in the prehabilitation group compared to the unmatched and matched standard

**Data Availability Statement:** All relevant data are within the paper and its Supporting Information files.

**Funding:** This study was financially supported by Edwards Lifesciences SA, Abbott (former St. Jude Medical Nederland B.V.) and 'Stichting Beatrixoord Noord-Nederland', who were not involved in the study design or execution. This work was supported by Edwards Lifesciences SA [grant number EV 102016 to JH], Abbott (formerly St. Jude Medical Nederland B.V.) [to JH], and "Stichting Beatrixoord Noord-Nederland" [grant number 210.178 to SD]. The funders had no role in study design, data collection and analysis, decision to publish, or preparation of the manuscript.

**Competing interests:** JH, SD, and MAM report grants from Edwards Lifesciences, SA, Abbott (former St. Jude Medical Nederland B.V.), and 'Stichting Beatrixoord Noord-Nederland'. MAM reports grants from AtriCure, Getinge and consultancy from LivaNova. The remaining authors have nothing to disclose. This does not alter our adherence to PLOS ONE policies on sharing data and materials. There are no patents, products in development or marketed products provided by Edwards Lifesciences, which are associated with this research.

care group (resp. 14.3% vs. 23.8%, P = 0.040 and 14.3% vs. 25.3%, P = 0.030). For the other complications, no between-group differences were found.

## Conclusions

Prehabilitation might be beneficial to prevent postoperative AF. Patients participated safely in prehabilitation and were not at higher risk for postoperative complications. However, well-powered randomized controlled trials are needed to confirm and deepen these results.

## Introduction

More than 15,000 patients with ischemic heart disease, which is the leading cause of death in Western countries, undergo a coronary artery bypass graft (CABG), valve, and/or aortic surgery in the Netherlands [1, 2]. Although mortality has decreased due to improvements in operative care, the risks of postoperative complications are still high. Approximately 30% of the procedures are complicated by arrhythmias [3], 33% by lung complications [4], and 26% by delirium [5]. The occurrence of postoperative complications hampers the recovery after surgery and can lead to a higher rate of mortality or loss of independence [6]. A decline in postoperative complications would thus reduce the patient burden and healthcare costs. Therefore, it is important to explore whether the risks of complications can be reduced before surgery.

It is well known that the risk of complications has been associated with the preoperative state of the patient. Several risk factors, such as age or type of surgery, are unchangeable. However, some lifestyle-related risk factors are modifiable. For example, physical inactivity (present in ~45% of the cardiac surgery patients) has been associated with an increase in atrial fibrillation (AF) and delirium after cardiac surgery [7, 8]. Furthermore, Hulzebos et al. (2003) [4] showed that recent smoking behavior and poor lung function were risk factors for postoperative pulmonary complications. Smoking was also associated with postoperative AF [6]. Poor preoperative nutrition status (present in ~80%) has also been associated with adverse surgery outcomes [9, 10]. These findings suggest that postoperative complications could be prevented by enhancing preoperative physical and mental status.

Cardiac rehabilitation (CR) has been suggested to improve the preoperative status of the patient and therefore be potentially effective in preventing postoperative complications. The aim of CR is "to influence favourably the underlying cause of cardiovascular disease, as well as to provide the best possible physical, mental and social conditions" [11]. However, the evidence for the benefits and risks of preoperative CR (prehabilitation) is scarce. Evidence is usually derived from small trials investigating the effect of a single-component prehabilitation program, which often consist of inspiratory muscle training (IMT), on pulmonary complications, length of stay, and mortality [12, 13]. A multidisciplinary prehabilitation program consisting of different components (e.g. whole body exercise, dietary, and psychological guidance) can be effective on multiple aspects and might therefore be more effective than a single-component program. Since very little is known about the unintended consequences and effects of such multidisciplinary program, it is relevant to explore the effects of a multidisciplinary prehabilitation program on pulmonary complications as well as on other postoperative complications, such as arrhythmias or delirium.

This explorative study was designed to analyze retrospectively the potential short-term postoperative benefits regarding in-hospital complications and unintended consequences of a standardized multidisciplinary prehabilitation program (including aerobic exercise) compared

to standard care. Although this is an exploratory study, based on the shown effects of preoperative IMT [12, 13], we expect a lower rate of in-hospital complications and no unintended consequences in patients who followed multidisciplinary prehabilitation.

## Materials and methods

### Study design, setting, and sample size

In this retrospective study, data were collected from 880 patients who underwent elective cardiac surgery at the University Medical Center Groningen (UMCG, the Netherlands) between January 2014 and April 2017. The Heart Center of the UMCG is one of the 17 Heart Centers in the Netherlands specialized in cardiac surgery. In addition to patients referred from the UMCG, patients from closely collaborating hospitals (located 5 to 60 kilometers from the UMCG) also underwent cardiac surgery in the UMCG. All admitted patients (regardless of whether they were referred externally or internally) received one preoperative consultation in the UMCG during the waiting period and followed the standard hospital protocol of the UMCG. However, in February 2015 a multidisciplinary preoperative and postoperative rehabilitation program (the Heart-ROCQ pilot program) was implemented in the UMCG for internally referred patients. This resulted in a subgroup (n = 91) that followed prehabilitation and a subgroup (n = 789) that did not receive any prehabilitation during the waiting time.

### Participants

Adults admitted for elective (i.e. no main stem stenosis, unstable angina pectoris, or progressive symptoms) coronary artery bypass graft (CABG), valve surgery, aortic surgery, or combined procedures were included. Patients were excluded when accepted for transcatheter aortic valve implantation, aortic dissection, aortic descending repair, Morrow procedure, or congenital procedure. Furthermore, patients that were registered as objecting to the use of their medical data were excluded. Patients who were not able or not motivated to follow prehabilitation or followed a divergent rehabilitation program (i.e. an adapted prehabilitation program without aerobic cycling or an early-initiated postoperative inpatient CR program) were also excluded.

### Intervention

Patients who followed the Heart-ROCQ pilot program in addition to the standard surgical care were referred to as the PRE-group. Fig 1 gives an overview of the Heart-ROCQ pilot program, which was conducted at the UMCG, Center for Rehabilitation (located 6 km from the UMCG hospital).

### Comparison

Before February 2015, no prehabilitation was offered at the UMCG or at the referring hospitals. Only after discharge from the UMCG, an outpatient CR was offered as standard care in the referral hospital [14]. Patients who did not have the option to follow prehabilitation (i.e. patients who underwent surgery before February 2015 or were referred from a peripheral hospital) and thus followed only the standard surgical care, were referred to as the Standard-Care (SC) group.

### Ethics and endpoint

The postoperative in-hospital complications investigated in this study are shown in Table 1. At the UMCG, delirium is diagnosed according to the criteria of the Diagnostic and Statistical

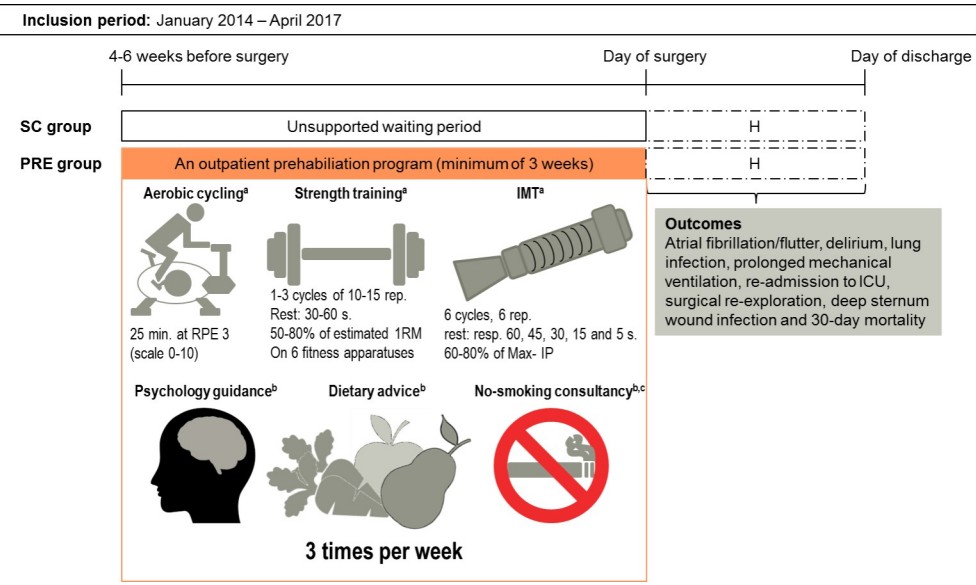

**Fig 1. An overview of the prehabilitation (PRE) group and standard care (SC) group.** The prehabilitation program was part of a pilot of the Heart-ROCQ program[28], which continued with a clinical inpatient rehabilitation phase following discharge from the hospital. Including two group education sessions about basic training principles and forced expiration technique and huff and cough techniques and 30 minutes sessions of body awareness were given every 2 weeks. [b]Including intake, individual sessions on indication, and group education on cardiovascular risk factors; [c]Only for patients who smoke; H: Hospitalization; ICU: Intensive care unit; IMT: Inspiratory muscle training; Max-IP: Peak inspiratory pressure; PRE-group: patients who followed the prehabilitation program; RPE: rate perceived exertion; SC-group: Standard Care group, who did not have the option to follow prehabilitation; 1RM: One Repetition Maximum, estimated based on 6-10RM.

**Table 1. Definitions of in-hospital postoperative complications.**

| Definitions outcomes |
| --- |
| Atrial fibrillation/ flutter |
| New onset of AF or atrial flutter on ECG or treatment of AF documented[a] |
| Delirium |
| Documentation of delirium diagnosis and treated with medication |
| Lung infection |
| Documentation of diagnosis of lung infection and medical treatment given |
| Prolonged mechanical ventilation |
| Mechanical ventilation longer than 24 hours during initial postoperative ICU stay |
| Re-admission to ICU |
| Surgical re-exploration |
| Surgical incision into the sternum as a result of a bleeding or tamponade |
| Deep sternum wound infection |
| Deeper tissues are affected (muscle, sternum, and mediastinum) and must include |
| 1) surgical drainage / re-fixation OR 2) an organism is isolated from culture of mediastina |
| tissue or fluid, OR 3) antibiotic treatment, because of sternum wound |
| 30-day mortality |
| All-cause mortality between the day of surgery and 30 days after surgery |

[a]Recurrence of AF/atrial flutter that was present preoperatively was not included; AF: Atrial fibrillation

ICU: Intensive care unit

Manual of Mental Disorders, 4<sup>th</sup> edition [15], and lung infections are diagnosed using the clinical definition of hospital-acquired pneumonia [16]. All patients received Beta-blockers postoperatively, unless they had conduction disturbances.

Major adverse clinical events (i.e. death, myocardial infarction, stroke, hospital readmission, change in surgery urgency, or sudden cardiac death) were evaluated during prehabilitation to investigate unintended consequences.

All collected data were de-identified according to the rules of the Dutch Privacy Law. A waiver for this study was granted by the Medical Ethics Review Board of the UMCG (METc UMC Groningen). Data were collected between March 2016 and February 2018 from existing databases (the Dutch registration database and the database of the intensive care unit) and medical records.

## Statistical analyses

Descriptive statistics included means (±SD) or median (interquartile range) for continuous variables and counts (percentages) for binary or categorical outcomes. Differences between the PRE-group and the SC-group were tested using the Pearson's chi-squared test. To reduce bias and improve balance between the groups, sensitivity analyses were performed using propensity-score matching [17, 18]. Before performing propensity-score matching, univariate analyses were conducted (IBM SPSS Statistics, version 23) on the baseline variables to identify which variables affected the outcomes. The variables that were significantly associated with the outcomes (Tables 2 and 3) were used in a logistic regression model to estimate the propensity score (i.e. the probability of treatment assignment) [19, 20]. The R package 'MatchIt' was used to perform Nearest-Neighbour-Matching. The matching procedure was performed without replacement and the "greedy" approach was used (i.e. it matched the closest control patient to the treated patient that had not yet been matched. The random order of selecting the treated patient affects the match process) [18–20]. In addition, different ratio's and caliper distances were used to find the optimal match. To find the optimal match, the balance of the baseline variables between the PRE-group and matched SC-group was checked by comparing the standardized mean differences and using Q-Q plots of the two empirical quantile functions (PRE- and SC-group) of each baseline variable [19]. The matched sample was most optimal (i.e. maximum balance, reduced heterogeneity and a large number of observations remaining) when no caliper distance and a ratio of 1:3 was used. The standardized difference of each baseline variable was less than 0.1 indicating a negligible difference between the PRE-group and matched SC-group [21] (S2 Appendix). The Pearson's chi-squared tests were performed using Stata (Version SE15.0). Level of significance was set at P<0.05 and statistical tests were all two-sided.

## Results

Between January 2014 and April 2017 1201 patients underwent elective cardiac surgery at the UMCG. In total 192 patients were excluded based on having had preoperative endocarditis, type of surgery, or being registered as objecting to the use of their medical data for scientific purposes (Fig 2). Of the patients, 97 were approached to follow prehabilitation, but were not motivated or able to follow it. Six patients followed an adapted program due to comorbidities and 26 patients followed an early-initiated postoperative inpatient CR program. Finally, 91 patients were in the PRE-group and 789 in the SC-group.

### Characteristics

The average duration of the prehabilitation program was 33.9±12.4 days. Tables 2 and 3 show preoperative and postoperative characteristics of the PRE-group and SC-group. Overall, the

**Table 2. Preoperative patient characteristics.**

| Preoperative characteristics | Unmatched groups | | | Groups matched by propensity score | | |
|---|---|---|---|---|---|---|
| | PRE-group (n = 91) | SC-group (n = 789) | P- value | PRE-group (n = 91) | SC-group (n = 273) | P-value |
| **Gender n (% men)**[a,b] | 71 (78.0%) | 549 (69.6%) | 0.095 | 71 (78.0%) | 206 (75.5%) | 0.620 |
| **Age (mean ± SD)**[a,c] | 64.5 (9.5) | 66.0 (9.8) | 0.210 | 64.5 (9.5) | 64.8 (9.8) | 0.840 |
| **BMI (mean ± SD)**[c] | 27.7 (4.6) | 27.6 (4.3) | 0.860 | 27.7 (4.6) | 27.7 (4.3) | 0.980 |
| **Left ventricular function**[a,d] | | | 0.560 | | | 1.000 |
| Poor LVEF (<31%) | 3 (3.3%) | 17 (2.2%) | | 3 (3.3%) | 8 (2.9%) | |
| Moderate LVEF (31–50%) | 24 (26.4%) | 191 (24.2%) | | 24 (26.4%) | 74 (27.1%) | |
| Good LVEF (>50%) | 64 (70.3%) | 581 (73.6%) | | 64 (70.3%) | 191 (70.0%) | |
| **NYHA-class**[a,d] | | | <0.001* | | | 0.620 |
| Class I | 6 (6.6%) | 41 (5.2%) | | 6 (6.6%) | 29 (10.6%) | |
| Class II | 58 (63.7%) | 333 (42.2%) | | 58 (63.7%) | 159 (58.2%) | |
| Class III | 24 (26.4%) | 388 (49.2%) | | 24 (26.4%) | 78 (28.6%) | |
| Class IV | 3 (3.3%) | 27 (3.4%) | | 3 (3.3%) | 7 (2.6%) | |
| **Logistic Euroscore II**[a,e] | 1.3 (0.9, 2.7) | 1.6 (1.0, 2.8) | 0.058 | 1.3 (0.9, 2.7) | 1.4 (0.9, 2.5) | 0.920 |
| **Waiting time (days)**[e,f] | 55 (51, 66) | 54 (36, 77) | 0.093 | 55 (51, 66) | 55 (36, 78) | 0.160 |
| **Chronic lung disease**[a,b] | 7 (7.7%) | 125 (15.8%) | 0.039* | 7 (7.7%) | 23 (8.4%) | 0.830 |
| **Diabetes mellitus**[b] | 20 (22.0%) | 166 (21.0%) | 0.840 | 20 (22.0%) | 57 (20.9%) | 0.820 |
| **Atrial fibrillation/flutter**[a,b] | 18 (19.8%) | 150 (19.0%) | 0.860 | 18 (19.8%) | 55 (20.1%) | 0.940 |
| **Recent myocardial infarct**[b] | 8 (8.8%) | 59 (7.5%) | 0.650 | 8 (8.8%) | 19 (7.0%) | 0.560 |
| **Previous PCI**[b] | 23 (25.3%) | 144 (18.2%) | 0.110 | 23 (25.3%) | 50 (18.3%) | 0.150 |
| **Previous cardiac surgery**[a,b] | 3 (3.3%) | 36 (4.6%) | 0.580 | 3 (3.3%) | 10 (3.7%) | 0.870 |
| **History of CVA**[a,b] | 5 (5.5%) | 48 (6.1%) | 0.820 | 5 (5.5%) | 15 (5.5%) | 1.000 |
| **From peripheral hospital**[b] | 3 (3.3%) | 603 (76.4%) | <0.001* | 3 (3.3%) | 201 (73.6%) | <0.001* |

Values are shown as median (Interquartile range) or n (% yes), unless otherwise noted.

[a] Variable significantly associated with the outcomes and included in the logistic regression model for the propensity score

[b] Pearson's chi-squared test

[c] Two sample t test

[d] Fisher's exact test

[e] Wilcoxon rank-sum test

[f] At time of acceptance for surgery till the day of surgery. BMI: Body mass index; CVA: Cerebral vascular accident; LVEF: Left ventricular ejection fraction; NYHA: New York Heart Association; PCI: Percutaneous coronary intervention; PRE-group: Prehabilitation group; SC-group: Standard care group SD: Standard deviation.

majority was male (70.5%), older than 65 years (60%) and overweight (72%, BMI>25.0kg/m2). Furthermore, 28% was obese (BMI>30.0kg/m2) and 20% suffered from diabetes. The prevalence of chronic lung disease and NYHA-class before surgery was significantly higher in the unmatched SC-group. In addition, there was a significant difference in complexity of surgery and on-pump and off-pump surgery between these groups. The PRE-group showed more isolated CABG interventions compared with more single valve interventions in the unmatched SC-group. Consequently, in the unmatched SC-group significantly more cardiopulmonary bypass surgeries were performed. Following the propensity analysis the balance of the baseline characteristics was improved between the PRE- and matched SC-group (Tables 2, 3, and S2 Appendix).

## Postoperative complications

Table 4 shows the incidence of the complications per group. AF was the most common complication, which was significantly lower in the PRE-group compared to the SC-group

**Table 3. Operative and postoperative characteristics.**

| Characteristics at time of hospitalization | Unmatched groups | | | Groups matched by propensity score | | |
|---|---|---|---|---|---|---|
| | PRE-group (n = 91) | SC-group (n = 789) | P- value | PRE-group (n = 91) | SC-group (n = 273) | P-value |
| **Complexity of surgery[a,c]** | | | 0.010* | | | 0.065 |
| **CABG isolated** | 47 (51.6%) | 313 (39.7%) | | 47 (51.6%) | 121 (44.3%) | |
| **Single, non CABG** | 16 (17.6%) | 256 (32.4%) | | 16 (17.6%) | 80 (29.3%) | |
| • Valve procedure | 15 (93.8)% | 208 (95.3%) | | 15 (93.8)% | 73 (91.3%) | |
| • Aorta surgery | 1 (6.3%) | 12 (4.7%) | | 1 (6.3%) | 7 (8.8%) | |
| **Two interventions** | 26 (28.6%) | 183 (23.2%) | | 26 (28.6%) | 58 (21.2%) | |
| **Three interventions** | 2 (2.2%) | 37 (4.7%) | | 2 (2.2%) | 14 (5.1%) | |
| **On-pump surgery[b]** | 52 (57.1%) | 552 (70.0%) | 0.013* | 52 (57.1%) | 180 (65.9%) | 0.130 |
| Bypass time (min)[d] | 149 (112, 181) | 135 (104, 180) | 0.240 | 149 (112, 181) | 136 (105, 176) | 0.300 |
| Cross clamp time (min)[d] | 97 (76, 132) | 90 (66, 126) | 0.220 | 97 (76, 132) | 90 (65, 124) | 0.220 |
| **Surgery time (min)[d]** | 219 (182, 269) | 217 (180, 272) | 0.780 | 219 (182, 269) | 215.0 (181, 260) | 0.680 |
| **Hospital Stay (days)[d]** | 6 (6, 10) | 7 (5, 8) | 0.370 | 6 (6, 10) | 7 (5, 9) | 0.610 |

Values are shown as median (Interquartile range) or n (% yes), unless otherwise noted.

[a] Variable significantly associated with the outcomes and included in the logistic regression model for the propensity score

[b] Pearson's chi-squared test

[c] Fisher's exact test

[d] Wilcoxon rank-sum test; CABG: Coronary artery bypass graft; PRE-group: Prehabilitation group; SC-group: Standard care group.

(P = 0.040 and P = 0.030 for resp. the unmatched and matched groups). The other complications were not significantly different between the groups.

## Safety

The surgery urgency changed from elective to urgent or emergent in three patients of the PRE-group (3.3%). This was comparable to the SC-group in which the surgery of 27 patients (3.4%) was moved forward. One patient was hospitalized during prehabilitation due to an infected wound seroma. The adverse events of the PRE-group did not occur during or directly after a prehabilitation session.

## Discussion

The aim of this study was to explore the potential short-term postoperative benefits and unintended consequences of prehabilitation with respect to in-hospital complications. Our analyses showed a significant decrease of AF in patients who followed prehabilitation compared to (matched) patients who did not follow prehabilitation. No unintended consequences or benefits of prehabilitation were shown regarding the other in-hospital complications and adverse events during prehabilitation.

To the best of our knowledge, this is the first study that has investigated the effects of a multidisciplinary prehabilitation program on postoperative complications such as AF and delirium. Preventing these complications would help to reduce mortality, morbidity and healthcare costs [6, 22–24]. The incidence of AF was almost 2-fold lower in the PRE-group compared with the SC-group. The development of postoperative AF has, regardless the use of a cardiopulmonary bypass, been associated with an increase of preoperative and postoperative inflammatory markers [3, 25]. Inflammation is therefore suggested as an important factor in the multifactorial pathophysiological mechanisms that cause postoperative AF [26]. On the other hand, the anti-inflammatory effects of exercise, healthy food, and psychological stress

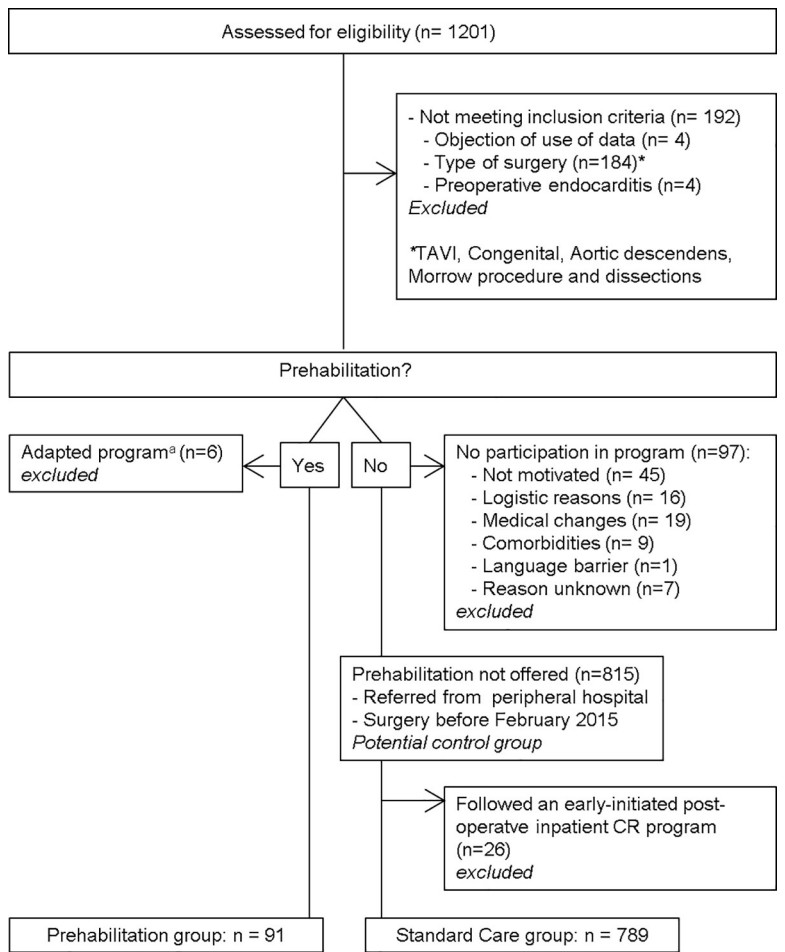

**Fig 2. Flow chart of the recruitment.** [a]Patients who were not able to cycle on an ergometer and followed other exercise modalities.

**Table 4. Incidence of in-hospital postoperative complications.**

| | Unmatched groups | | | Groups matched by propensity score | | |
|---|---|---|---|---|---|---|
| Outcomes | PRE-group (n = 91] | SC-group (n = 789) | P-value[a] | PRE-group (n = 91) | SC-group (n = 273) | P-value[a] |
| **Atrial fibrillation/flutter** | 13 (14.3%) | 188 (23.8%) | 0.040* | 13 (14.3%) | 69 (25.3%) | 0.030* |
| **Delirium** | 14 (15.4%) | 96 (12.2%) | 0.380 | 14 (15.4%) | 27 (9.9%) | 0.150 |
| **Lung infection** | 6 (6.6%) | 40 (5.1%) | 0.540 | 6 (6.6%) | 16 (5.9%) | 0.800 |
| **Prolonged MV** | 8 (8.8%)[b] | 35 (4.4%)[c] | 0.068 | 8 (8.8%)[b] | 14 (5.1%)[c] | 0.200 |
| **Re-admission ICU** | 9 (9.9%) | 46 (5.8%) | 0.130 | 9 (9.9%) | 16 (5.9%) | 0.190 |
| **Surgical re-exploration** | 7 (7.7%) | 39 (4.9%) | 0.260 | 7 (7.7%) | 11 (4.0%) | 0.160 |
| **Deep sternum wound infection** | 1 (1.1%) | 11 (1.4%) | 0.820 | 1 (1.1%) | 6 (2.2%) | 0.510 |
| **30-day mortality** | 2 (2.2%) | 9 (1.1%) | 0.390 | 2 (2.2%) | 2 (0.7%) | 0.250 |

Values are shown as n (% yes).

[a]Pearson's chi-squared test

[b] median duration of 7.0 (IQR 4.5–11.0) hours

[c] median duration of 7.0 (IQR 5.0–10.0) hours; ICU: Intensive care unit; MV: Mechanical ventilation.

reduction (offered to the PRE-group) are well stated [27–29]. Therefore, it would be interesting to research whether there might be a causal relation between the possible anti-inflammatory effects of the Heart-ROCQ-pilot program on the pre- and postoperative inflammatory state and the risks of AF.

Regarding the other postoperative complications, there were no significant differences, which lead to the assumption that the prehabilitation program (including aerobic exercise) was safe. However, the definition of postoperative complications was based on clinical findings and routine testing instead of systematic monitoring. Therefore, the incidence of postoperative complications might have been underestimated, which resulted, together with a limited sample size of the PRE-group, in a lower power of the study. Similar results were found regarding PMV and all-cause mortality [12]. In contrast to other studies including high-risk patients, our study, which included an all-comers sample, found no reduction in lung infection. Lung infection was also not reduced after prehabilitation in surgical lung cancer patients, although other pulmonary complications were decreased [30]. The content and duration of prehabilitation were different in these studies. Cardiac patients in other studies received inspiratory muscle training (IMT) on a daily basis, whereas patients in our study received a multidisciplinary prehabilitation program (including IMT) for three times per week over a longer period. In general short (1–2 weeks) and intensive (3–10 times per week) programs were offered to surgical lung cancer patients resulting in preoperative physical improvement and a shorter hospital stay[30]. However, it is still unclear which complications can be prevented by this physical improvement. Also, the feasibility and trainability of these programs with higher exercise intensity is unknown in cardiac patients. Results on the feasibility and trainability of our prehabilitation program are submitted.

A strength of this retrospective study design is that it includes an all-comers population, which are observed in a real-life setting. However, a limitation was that patients were not randomized. Patients with a higher NYHA-class and chronic lung disease were less often referred to prehabilitation. Possibly, these patients were operated with higher priority. Furthermore, patients from the PRE-group were mainly referred from the UMCG, while patients from the SC-group were mainly referred from peripheral hospitals. However, the referring hospital was not associated with postoperative outcomes (S1 Appendix). Propensity-score matching was used to minimize bias and to improve the balance between the groups [18]. Patients from 2014 were also included resulting in a larger control group. This gave the advantage that each treated patient could be matched to three control patients, resulting in an increased precision of the match and a reduction in the bias [19]. In a re-analysis of patients included from 2015 and onwards, the same trends were found (S3 Appendix). Although we created well-balanced groups, we cannot rule out the possibility of a certain degree of selection bias or unmeasured confounding factors that contributed to our findings. We hope the future results of our on-going study will serve as a sufficiently and carefully organized study that rules out the elements of bias and confounding factors [31].

In conclusion, prehabilitation (including aerobic exercise) is a promising therapy, it is inexpensive and easy to implement, since postoperative rehabilitation is already implemented as standard care. It might be beneficial to prevent postoperative AF and patients were not at higher risk for postoperative complications, suggesting the program is safe. Well-powered randomized controlled trials are needed to confirm these findings and further explore the preoperative and (short- and long-term) postoperative effects of prehabilitation and of who will benefit from prehabilitation.

## Supporting information

**S1 Appendix. Univariate analyses.**
(DOCX)

**S2 Appendix. Balance between matched and unmatched data.**
(DOCX)

**S3 Appendix. Re-analysis of data from 2015 and onwards.**
(DOCX)

## Acknowledgments

We thank Willem Dieperink and Tjalling Waterbolk for their help with the data collection. Mike DeJongste is thanked for his support in the study.

## Author Contributions

**Conceptualization:** Johanneke Hartog, Iman Mousavi, Pim van der Harst, Massimo A. Mariani.

**Data curation:** Johanneke Hartog, Iman Mousavi.

**Formal analysis:** Johanneke Hartog.

**Funding acquisition:** Johanneke Hartog.

**Methodology:** Johanneke Hartog, Iman Mousavi, Pim van der Harst, Massimo A. Mariani.

**Supervision:** Joke Fleer, Lucas H. V. van der Woude, Pim van der Harst, Massimo A. Mariani.

**Visualization:** Johanneke Hartog.

**Writing – original draft:** Johanneke Hartog.

**Writing – review & editing:** Johanneke Hartog, Iman Mousavi, Sandra Dijkstra, Joke Fleer, Lucas H. V. van der Woude, Pim van der Harst, Massimo A. Mariani.

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
