## [Decision Letter · Decision Letter 0]

2 Feb 2021

PONE-D-20-40737

Prehabilitation to prevent complications after cardiac surgery - A retrospective study

PLOS ONE

Dear Dr. Hartog,

Thank you for submitting your manuscript to PLOS ONE. After careful consideration, we feel that it has merit but does not fully meet PLOS ONE’s publication criteria as it currently stands. Therefore, we invite you to submit a revised version of the manuscript that addresses the points raised during the review process.

Please address the issues and revise accordingly.

We look forward to receiving your revised manuscript.

Kind regards,

Academic Editor

PLOS ONE

Journal Requirements:

2. In the ethics statement in the manuscript and in the online submission form, please provide additional information about the patient records/samples used in your retrospective study, including the date range (month and year) during which patients' medical records/samples were accessed.

4.We note that the grant information you provided in the ‘Funding Information’ and ‘Financial Disclosure’ sections do not match.

5.Thank you for stating the following in the Competing Interests section:

"JH, SD, and MAM report grants from Edwards Lifesciences, SA, Abbott (former St. Jude Medical Nederland B.V.), and ‘Stichting Beatrixoord Noord-Nederland’. MAM reports grants from AtriCure, Getinge and consultancy from LivaNova. The remaining authors have nothing to disclose."

Reviewers' comments:

Reviewer's Responses to Questions

**Comments to the Author**

1. Is the manuscript technically sound, and do the data support the conclusions?

Reviewer #1: Yes

Reviewer #2: No

2. Has the statistical analysis been performed appropriately and rigorously? 

Reviewer #1: Yes

Reviewer #2: Yes

3. Have the authors made all data underlying the findings in their manuscript fully available?

Reviewer #1: Yes

Reviewer #2: Yes

4. Is the manuscript presented in an intelligible fashion and written in standard English?

Reviewer #1: Yes

Reviewer #2: No

5. Review Comments to the Author

Reviewer #1: Dear the authors of the manuscript entitled "Prehabilitation to prevent complications after cardiac surgery - A retrospective study"

Thank you for doing this study

I think this study is an excellent one since it highlights the importance of preoperative rehabilitation program in decreasing the incidence of postoperative complications after elective cardiac surgery

This study demonstrated that there was clear benefit of exercise programs in in decreasing the incidence of atrial fibrillation after cardiac surgery, however it did not show a clear benefit in improving the other outcomes

there are some limitations of this study and there were mentioned by the authors, however I think this study had good power since it included all comers and had propensity matched groups

All in all I have no concerns about this manuscript

Thank you

Reviewer #2: This study is a single-center, retrospective study that aims to explore the impact of a prehabilitation program on short-term outcomes following cardiac surgery. This is an interesting study of ultimately 364 patients spanning between 2014 to 2017. The authors found that patients that underwent prehabilitation had a lower incidence of atrial fibrillation (14.3% vs 25.3%) compared to the standard of care. Comments and questions are enumerated.

1. A major limitation of this study is the lack of randomization of the two treatment arms. This was addressed in the discussion section but may benefit from earlier introduction in the methods section. Propensity matching was utilized to account for patient and hospital differences but unfortunately this does not account for bias such as assessment of patient functional status.

2. Did the authors run a power analysis? The study’s sample size was very small for both treatment arms after propensity matching. The outcomes of interest such as mortality, sternum wound infection, lung infection and surgical re-exploration were rare events and may have not occurred enough to run a robust analysis. Characteristics such as patients who underwent aortic surgery were also rare events.

3. Were the authors able to run a sub-analysis of the patients that experienced postoperative atrial fibrillation? While the authors proposed that inflammation may be a major driver of the difference in this complication, it would be interesting to see how functional status, comorbidities, complexity of surgery, cross clamp and bypass times compared in these patients.

4. Minor: The Diagnostic and Statistical Manual of Mental Disorders 5th edition is the current standard.

6. PLOS authors have the option to publish the peer review history of their article (what does this mean?). If published, this will include your full peer review and any attached files.

Reviewer #1: **Yes: **Salah Eldien Altarabsheh

Reviewer #2: No

---

## [Author Response · Author response to Decision Letter 0]

7 Apr 2021

University Medical Center Groningen 

Hanzeplein 1, AB41, 

9713 GZ Groningen 

The Netherlands 

Phone:+3150 3617527 

Email: j.hartog@umcg.nl

Robert Jeenchen Chen, MD, MPH 16 March 2021

Academic Editor

PLOS ONE

Email: plosone@plos.org. 

Dear Professor Robert Jeenchen Chen:

On behalf of the authors, I thank you and the reviewers for the time and effort for examining our manuscript. Furthermore, we are thankful for the opportunity to submit a revised version of our manuscript “Prehabilitation to prevent complications after cardiac surgery - A retrospective study with propensity score analysis”. Since data and knowledge about the effect of prehabilitation in cardiac surgical patients is scarce, we think this retrospective analysis is a first and valuable indication of the safety and possible potential of prehabilitation in open heart surgery patients and valuable to share with the readership of PLOS ONE.

Based on your and the reviewers’ comments, the manuscript is adapted and improved. We used the track changes mode in MS Word to highlight the changes made in the manuscript. Please find attached our detailed response to the reviewers’ comments. Page numbers in this response refer to the version of the manuscript with track changes. 

All authors have contributed significantly and have read and approved the submission of this revised manuscript. We hope our revised manuscript is suitable for publication in PLOS ONE.

On behalf of all authors,

Yours sincerely,

Johanneke Hartog, MSc

PhD-candidate Cardiology and Thoracic surgery

PONE-D-20-40737 - Prehabilitation to prevent complications after cardiac surgery - A retrospective study PLOS ONE

Journal Requirements:

 Reply: Thank you for the observation. We changed the following aspects:

- The title page has been adapted to the PLOS ONE’s style requirements.

- The font style has been changed to ‘Italics’

- Level 1 and 2 headings are used for the headers throughout the manuscript in respectively ‘Italics, Bold typ, 18pt font’ and ‘Italics, Bold typ, 16pt font’.

2. In the ethics statement in the manuscript and in the online submission form, please provide additional information about the patient records/samples used in your retrospective study, including the date range (month and year) during which patients' medical records/samples were accessed.

 Reply: We added in the Materials and methods section ‘Data were collected between March 2016 and February 2018 from existing databases (the Dutch registration database and the database of the intensive care unit) and medical records.’ (p.5, line 88-89). We removed the sentence ‘Data were collected from existing databases and medical records.’ from the sub header ‘outcomes’ in the Materials and methods section (p.6, line 121, 122) 

 Reply: Thank you for your notice: 

- The section ‘Supporting Information’ has been added to the end of the manuscript, including the S1-S3 Appendix. The filenames of the supporting information files are changed into ‘S1 Appendix - Univariate analyses’, ‘S2 Appendix – Balance between matched and unmatched data’, and ‘S3 Appendix – Re-analysis of data from 2015 and onwards’ (p.17, line 336-339).

- The citations in the text are updated as well. 

4.We note that the grant information you provided in the ‘Funding Information’ and ‘Financial Disclosure’ sections do not match.

 Reply: We have adjusted the ‘Funding information’ so that it matched with ‘Financial Disclosure’ now. In addition, we add the sentence ‘The funders had no role in study design, data collection and analysis, decision to publish, or preparation of the manuscript.’ to the Financial Closure Statement section in the manuscript (p.17, line 328-330).

 Reply: We provided the correct grant numbers for Edwards Lifesciences and ‘Stichting Beatrixoord Noord-Nederland’. The grant number for SA, Abbott (former St. Jude Medical Nederland B.V.) is not applicable.

5.Thank you for stating the following in the Competing Interests section:

"JH, SD, and MAM report grants from Edwards Lifesciences, SA, Abbott (former St. Jude Medical Nederland B.V.), and ‘Stichting Beatrixoord Noord-Nederland’. MAM reports grants from AtriCure, Getinge and consultancy from LivaNova. The remaining authors have nothing to disclose."

 Reply: The sentence ‘This does not alter our adherence to PLOS ONE policies on sharing data and materials’ is added to the Competing Interests section (p.17, line 331-335).

Reviewer's Responses to Questions

Comments to the Author

1. Is the manuscript technically sound, and do the data support the conclusions?

Reviewer #1: Yes

Reviewer #2: No

2. Has the statistical analysis been performed appropriately and rigorously? 

Reviewer #1: Yes

Reviewer #2: Yes

3. Have the authors made all data underlying the findings in their manuscript fully available?

Reviewer #1: Yes

Reviewer #2: Yes

4. Is the manuscript presented in an intelligible fashion and written in standard English?

Reviewer #1: Yes

Reviewer #2: No

Reply: Thank you for this observation. The manuscript has been reviewed on textual grammar (textual changes are highlighted in yellow).

5. Review Comments to the Author

Reviewer #1: 

Dear the authors of the manuscript entitled "Prehabilitation to prevent complications after cardiac surgery - A retrospective study"

Thank you for doing this study

I think this study is an excellent one since it highlights the importance of preoperative rehabilitation program in decreasing the incidence of postoperative complications after elective cardiac surgery

This study demonstrated that there was clear benefit of exercise programs in in decreasing the incidence of atrial fibrillation after cardiac surgery, however it did not show a clear benefit in improving the other outcomes

there are some limitations of this study and there were mentioned by the authors, however I think this study had good power since it included all comers and had propensity matched groups

All in all I have no concerns about this manuscript

Thank you

Reply: Thank you for taking the time to review our manuscript and for the positive words about this study. We agree with you about the importance of preoperative rehabilitation. We hope that the results of this study will help to increase the awareness and understanding of the effects of prehabilitation and that it will encouraged clinicians and scientists to take the possibility of prehabilitation into account. Especially, now the risk factors, such as age, physical inactivity, obesity, diabetes, hypertension, and dyslipidaemia, are steadily increasing in patients undergoing cardiac surgery[1,2].

Reviewer #2: This study is a single-center, retrospective study that aims to explore the impact of a prehabilitation program on short-term outcomes following cardiac surgery. This is an interesting study of ultimately 364 patients spanning between 2014 to 2017. The authors found that patients that underwent prehabilitation had a lower incidence of atrial fibrillation (14.3% vs 25.3%) compared to the standard of care. Comments and questions are enumerated.

Reply: Thank you for your interest in this study and for your effort and time examining the manuscript. Please find below the answers of your comments and questions.

1. A major limitation of this study is the lack of randomization of the two treatment arms. This was addressed in the discussion section but may benefit from earlier introduction in the methods section. Propensity matching was utilized to account for patient and hospital differences but unfortunately this does not account for bias such as assessment of patient functional status.

Reply (response to the first part of the question): We agree that randomized controlled trials are preferred over observational studies to evaluate a new intervention, but observational designs have also their strengths. For example, an observational study includes an all-comers population in a natural setting. In addition, it’s explorative nature, though less controlled and powerful, helps to develop theory and potential future hypotheses. We used this study as an explorative study for eventually setting up such a randomized controlled trial to further systematically investigate the effects of prehabilitation on functional status, postoperative surgical complications, major adverse cardiac events, cardiorespiratory fitness, muscle strength, and mental health in a randomized and controlled manner[3]. Since data and knowledge about the effect of prehabilitation in cardiac surgical patients is scarce, we think this retrospective analysis is a first and valuable indication of the safety and possible potential of prehabilitation in open heart surgery patients and valuable to share with the readership of PLOS ONE. 

Changes in manuscript:

Thank you for your suggestion, we added ‘with propensity score analysis’ in the title. In this way, the reader is informed early about the design of this study. In our opinion, limitations should be addressed in the Discussion section, therefore we did not remove this limitation from the Discussion section. 

Reply (response to second part of the question):

Unfortunately, in this retrospective, exploratory study only outcomes that were collected in the standard care were available. However, patients who are physically constrained by their underlying cardiac condition are often operated with higher priority. By focussing on elective patients, all patients were able to carry out activities of daily living, independently . 

Although we do agree that there might be confounding factors that could have contributed to our findings, we think that we minimized these effects by using propensity score matching procedure. As advised in the literature[4]; we matched more patients from the SC group to each patient from the PRE group. Due to the considerably larger control group it was possible to find three comparable patients from the SC group for each individual patient in the PRE group. As a result, all baseline variables that potentially influenced the outcomes were well balanced between the matched SC group and the PRE group after this propensity score matching. As such, we think we chose an appropriate method to minimize the potential bias in this retrospective observational study, but, as we stated in the Discussion section, the current results should be confirmed by randomized controlled trials. 

2. Did the authors run a power analysis? The study’s sample size was very small for both treatment arms after propensity matching. The outcomes of interest such as mortality, sternum wound infection, lung infection and surgical re-exploration were rare events and may have not occurred enough to run a robust analysis. Characteristics such as patients who underwent aortic surgery were also rare events.

Reply: Thank you for your question. Because of the retrospective nature, we did not perform a power analysis. Indeed results of this study should be interpreted with caution, therefore we stated in the Discussion section that these results should be confirmed by well-powered randomized controlled studies (p.13, line 241). This study was however done in preparation to set up an actual prospective randomized controlled trial[3], with among others the current data we performed a power analysis for this prospective randomized controlled trial. The group who underwent aortic surgery is not well represented in the study, because it is a less common operation.

Changes to manuscript:

- ‘of the study’ has been added to the third paragraph in the Discussion section (p.12, line 213).

- ‘Randomized controlled trials’ has been added to the last paragraph in the Discussion section (p.13, line 241).

3. Were the authors able to run a sub-analysis of the patients that experienced postoperative atrial fibrillation? While the authors proposed that inflammation may be a major driver of the difference in this complication, it would be interesting to see how functional status, comorbidities, complexity of surgery, cross clamp and bypass times compared in these patients.

Reply: We thank the reviewer for the interesting question. If we interpret this question correctly, the reviewer suggests to investigate whether there are other factors besides inflammation that may influence the occurrence of atrial fibrillation. A comparison between the patients with postoperative atrial fibrillation (POAF) of the PRE-group and the patients with POAF of the SC-group for the variables, functional status, comorbidities, complexity of surgery, cross clamp and bypass times, is suggested as sub-analysis.

Indeed, as stated in the Discussion section (p.11, line 201,202) the mechanisms resulting in POAF are multifactorial[5]. Univariate analyses, which we conducted in preparation of the propensity analysis, showed that factors such as chronic lung disease and previous cardiac surgery were (borderline) predictors of POAF, while NYHA class and the complexity of the surgery were no predictors of POAF (S1 Appendix – univariate analyses). Furthermore, we agree with the reviewer that the use and duration of cardiopulmonary bypass can up-regulate the body’s inflammatory response to heart surgery resulting in a higher incidence of POAF. Since a sub-analysis on patients with POAF would downgrade the study power (since the sample size for the PRE-group only will be 13 patients), we have chosen to perform a propensity-score matching analysis. After matching the standardized differences, all above mentioned variables were less than 0.1 indicating a negligible difference between the PRE-group and matched SC-group. Despite the good balance in these variables between the groups, there was still a significant difference on POAF, assuming that prehabilitation is also contributing in reducing the incidence of POAF. 

It is well stated that, besides postoperative inflammatory markers, preoperative inflammatory markers are also risk factors in the development of POAF, regardless the use of a cardiopulmonary bypass[5-8]. On the other hand, the anti-inflammatory effects of exercise, healthy food, and stress reduction are well known[9-11]. Therefore, we find it interesting to research whether there might be a causal relation between the possible anti-inflammatory effects of the Heart-ROCQ-pilot program on the pre- and postoperative inflammatory state and the risks of AF. 

Changes in manuscript:

- The sentences ‘The development of postoperative AF has been associated with an increase of pre- and postoperative inflammatory markers[1]. It is likely that inflammation is an important factor in the multifactorial pathophysiological mechanisms that cause postoperative AF. It is possible that exercise, healthy food, and psychological stress reduction (offered to the PRE-group) down-regulated the preoperative inflammatory state, whereby the inflammatory response from cardiac surgery was modulated, reducing the risk of AF[23–26]. However, these possible mechanisms should be investigated further.’ are changed into ‘The development of postoperative AF has, regardless the use of a cardiopulmonary bypass, been associated with an increase of preoperative and postoperative inflammatory markers[1, 23]. Inflammation is therefore suggested as an important factor in the multifactorial pathophysiological mechanisms that cause postoperative AF[24]. On the other hand, the anti-inflammatory effects of exercise, healthy food, and psychological stress reduction (offered to the PRE-group) are well stated[25–27]. Therefore, it would be interesting to research whether there might be a causal relation between the possible anti-inflammatory effects of the Heart-ROCQ-pilot program on the pre- and postoperative inflammatory state and the risks of AF.’ in the second paragraph of the Discussion section (p.12,13, line 199-208)

4. Minor: The Diagnostic and Statistical Manual of Mental Disorders 5th edition is the current standard.

Reply: We agree with the reviewer. In the University Medical Center Groningen, delirium is currently diagnosed according to the criteria of the Diagnostic and Statistical Manual of Mental Disorders 5th edition.

6. PLOS authors have the option to publish the peer review history of their article (what does this mean?). If published, this will include your full peer review and any attached files.

Do you want your identity to be public for this peer review? For information about this choice, including consent withdrawal, please see our Privacy Policy.

Reviewer #1: Yes: Salah Eldien Altarabsheh

Reviewer #2: No

References

1 Ghanta RK, Kaneko T, Gammie JS, et al. Evolving trends of reoperative coronary artery bypass grafting: An analysis of the Society of Thoracic Surgeons Adult Cardiac Surgery Database. J Thorac Cardiovasc Surg 2013;145:364–372.

2 Kindo M, Hoang Minh T, Perrier S, et al. Trends in isolated coronary artery bypass grafting over the last decade. Interact Cardiovasc Thorac Surg 2017;24:71–76.

3 Hartog J, Blokzijl F, Dijkstra S, et al. Heart Rehabilitation in patients awaiting Open heart surgery targeting to prevent Complications and to improve Quality of life (Heart-ROCQ): study protocol for a prospective, randomised, open, blinded endpoint (PROBE) trial 2019;9:e031738.

4 Deb S, Austin PC, Tu J V., et al. A Review of Propensity-Score Methods and Their Use in Cardiovascular Research. Can J Cardiol 2016;32:259–265.

5 Boons J, Van Biesen S, Fivez T, et al. Mechanisms, Prevention, and Treatment of Atrial Fibrillation After Cardiac Surgery: A Narrative Review. J Cardiothorac Vasc Anesth 2020;000:1–10.

6 Lo B, Fijnheer R, Nierich AP, et al. C-reactive protein is a risk indicator for atrial fibrillation after myocardial revascularization. Ann Thorac Surg 2005;79:1530–1535.

7 Weymann A, Popov AF, Sabashnikov A, et al. Baseline and postoperative levels of C-reactive protein and interleukins as inflammatory predictors of atrial fibrillation following cardiac surgery: a systematic review and meta-analysis. Kardiol Pol 2018;76:440–451.

8 Todorov H, Janssen I, Honndorf S, et al. Clinical significance and risk factors for new onset and recurring atrial fibrillation following cardiac surgery - a retrospective data analysis. BMC Anesthesiol 2017;17:1–10.

9 Giugliano D, Ceriello A, Esposito K. The Effects of Diet on Inflammation. Emphasis on the Metabolic Syndrome. J Am Coll Cardiol 2006;48:677–685.

10 Zanchi NE, Almeida FN, Lira FS, et al. Renewed avenues through exercise muscle contractility and inflammatory status. ScientificWorldJournal 2012;2012:1–7.

11 Wirtz PH, von Känel R. Psychological Stress, Inflammation, and Coronary Heart Disease. Curr Cardiol Rep 2017;19:1–10.

---

## [Decision Letter · Decision Letter 1]

11 Apr 2021

PONE-D-20-40737R1

Prehabilitation to prevent complications after cardiac surgery - A retrospective study with propensity score analysis

PLOS ONE

Dear Dr. Hartog,

Thank you for submitting your manuscript to PLOS ONE. After careful consideration, we feel that it has merit but does not fully meet PLOS ONE’s publication criteria as it currently stands. Therefore, we invite you to submit a revised version of the manuscript that addresses the points raised during the review process.

Please revise accordingly.

We look forward to receiving your revised manuscript.

Kind regards,

Academic Editor

PLOS ONE

Journal Requirements:

Reviewers' comments:

Reviewer's Responses to Questions

**Comments to the Author**

1. If the authors have adequately addressed your comments raised in a previous round of review and you feel that this manuscript is now acceptable for publication, you may indicate that here to bypass the “Comments to the Author” section, enter your conflict of interest statement in the “Confidential to Editor” section, and submit your "Accept" recommendation.

Reviewer #1: All comments have been addressed

Reviewer #3: All comments have been addressed

2. Is the manuscript technically sound, and do the data support the conclusions?

Reviewer #1: Yes

Reviewer #3: Partly

3. Has the statistical analysis been performed appropriately and rigorously? 

Reviewer #1: Yes

Reviewer #3: Yes

4. Have the authors made all data underlying the findings in their manuscript fully available?

Reviewer #1: Yes

Reviewer #3: Yes

5. Is the manuscript presented in an intelligible fashion and written in standard English?

Reviewer #1: Yes

Reviewer #3: Yes

6. Review Comments to the Author

Reviewer #1: Dear the authors

Thank you again for taking all considerations for the reviewers comments

I have no concerns about this manuscript

Reviewer #3: Introduction:

- The introduction does not cover all the elements of the study.

- I suggest the authors expose to the following points in the introduction: What is known about the cardiac surgery? What is not known? Why the study was done?".

- Also, add the hypothesis of the study.

Methods:

- The methods cannot be followed in this frame.

- Kindly, re-frame the methods in accordance with components (SPICES) for methods:

a. Study design, setting, sample size

b. Participant

c. Intervention/issue of interest (exposure)

d. Comparison

e. Ethics and endpoint

f. Statistical analysis

Discussion:

Kindly, reframe it as the following:

a. Main findings of the present study

b. Comparison with other studies

c. Implication, and explanation of findings

d. Strengths and limitations

e. Conclusion, recommendation and future direction.

7. PLOS authors have the option to publish the peer review history of their article (what does this mean?). If published, this will include your full peer review and any attached files.

Reviewer #1: **Yes: **Salah Altarabsheh

Reviewer #3: **Yes: **Walid Kamal Abdelbasset

---

## [Author Response · Author response to Decision Letter 1]

27 May 2021

University Medical Center Groningen 

Hanzeplein 1, AB41, 

9713 GZ Groningen 

The Netherlands 

Phone:+3150 3617527 

Email: j.hartog@umcg.nl

Robert Jeenchen Chen, MD, MPH 25 May 2021

Academic Editor

PLOS ONE

Email: plosone@plos.org. 

Dear Professor Robert Jeenchen Chen:

Thank you for the opportunity to submit a second revised version of our manuscript “Prehabilitation to prevent complications after cardiac surgery - A retrospective study with propensity score analysis”. 

Based on the reviewers’ comments, the manuscript has been adapted. We used the track changes mode in MS Word to highlight the changes made in the manuscript. Please find attached our detailed response to the reviewers’ comments. Page numbers in this response refer to the version of the manuscript with track changes. 

We think that our manuscript has improved considerably. All authors have contributed significantly and have read and approved the submission of this revised manuscript. We hope our revised manuscript is suitable for publication in PLOS ONE.

On behalf of all authors,

Yours sincerely,

Johanneke Hartog, MSc

PhD-candidate Cardiology and Thoracic surgery

 

PONE-D-20-40737R1 - Prehabilitation to prevent complications after cardiac surgery - A retrospective study with propensity score analysis

Reviewers' comments:

Reviewer's Responses to Questions

Comments to the Author

1. If the authors have adequately addressed your comments raised in a previous round of review and you feel that this manuscript is now acceptable for publication, you may indicate that here to bypass the “Comments to the Author” section, enter your conflict of interest statement in the “Confidential to Editor” section, and submit your "Accept" recommendation.

Reviewer #1: All comments have been addressed

Reviewer #3: All comments have been addressed

2. Is the manuscript technically sound, and do the data support the conclusions?

Reviewer #1: Yes

Reviewer #3: Partly

3. Has the statistical analysis been performed appropriately and rigorously? 

Reviewer #1: Yes

Reviewer #3: Yes

4. Have the authors made all data underlying the findings in their manuscript fully available?

Reviewer #1: Yes

Reviewer #3: Yes

5. Is the manuscript presented in an intelligible fashion and written in standard English?

Reviewer #1: Yes

Reviewer #3: Yes

6. Review Comments to the Author

Reviewer #1: Dear the authors

Thank you again for taking all considerations for the reviewers comments

I have no concerns about this manuscript

Reply: Thank you for your effort to review our manuscript.

Reviewer #3: Introduction:

- The introduction does not cover all the elements of the study.

- I suggest the authors expose to the following points in the introduction: What is known about the cardiac surgery? What is not known? Why the study was done?".

Reply: Thank you for this suggestion. We reviewed the introduction thoughtfully, please find our comments and revisions regarding your suggestions below. 

What is known about the cardiac surgery

In the first and second paragraph we focus mainly on what is known about cardiac surgery with respect to what is relevant to our research question. It describes the impact of the surgery nowadays (first paragraph) and how the risk on postoperative complications is linked to the patients’ preoperative status (second paragraph). 

Changes to manuscript:

- To give more background information about cardiac surgery we added ‘More than 15,000 patients with ischemic heart disease, which is the leading cause of death in Western countries, undergo a coronary artery bypass graft (CABG), valve, and/or aortic surgery in the Netherlands[1,2].’ (Paragraph 1, p.3, line: 46-47) and the second sentence was changed from ‘Cardiac surgery is a major intervention, and despite the recent improvements in operative care, the risks of postoperative complications are still high.’ into ‘Although mortality has decreased due to improvements in operative care, the risks of postoperative complications are still high.’ (paragraph 1, p.3, line: 47-48). 

- In addition, ‘it is well known that’ is added to the first sentence of the second paragraph to emphasize that this paragraph is about what is already known (p.3, line: 55).

What is not known

In the third paragraph we indicated the current knowledge gap among the effects of prehabilitation in cardiac surgical patients. It is stated that the literature mainly consists of small trials, which are mainly focused on inspiratory muscle training as prehabilitation program. In addition, it is stated that the effects of a multidisciplinary prehabilitation program remain unclear. 

Changes to manuscript:

- To clarify this point we changed the sentences ‘However, the evidence for the benefits and risks of multidisciplinary --- (e.g. arrhythmias or delirium) remain unclear.’ into ‘However, the evidence for the benefits and risks of preoperative CR (prehabilitation) is scarce. Evidence is usually derived from small trials investigating the effect of a single-component prehabilitation program, which often consist of inspiratory muscle training (IMT), on pulmonary complications, length of stay, and mortality[10,11]. A multidisciplinary prehabilitation program consisting of different components (e.g. whole body exercise, dietary’, and psychological guidance) can be effective on multiple aspects and might therefore be more effective than a single-component program. Since very little is known about the unintended consequences and effects of such multidisciplinary program, it is relevant to explore the effects of a multidisciplinary prehabilitation program on pulmonary complications as well as on other postoperative complications such as arrhythmias or delirium.’ (Third paragraph, p.3-4, line: 63-75).

Why the study was done

We emphasized the importance of this study by mentioning the high number of postoperative complications and these consequences in the first paragraph (p.3, line: 49-52) and mention the association between preoperative status and the risk of postoperative complications (Paragraph 2, P.3, line: 55-62). To accentuate the aspect of why the study was done, we did the below mentioned revisions. 

Changes to manuscript:

- We added the sentence ‘A decline in postoperative complications would thus reduce the patient burden and healthcare costs. Therefore, it is important to explore whether the risks of complications can be reduced before surgery.’ (p.3, line: 53-54). 

- We moved the reason why cardiac rehabilitation or prehabilitation is suggested forward into the third paragraph (p.3, line: 63-66). The first sentence of this paragraph is thus changed from ‘Cardiac rehabilitation (CR) aims “to influence favourably the underlying cause of cardiovascular disease, as well as to provide the best possible physical, mental and social conditions”[9] and therefore has been suggested to improve the preoperative status of the patient.’ into ‘Cardiac rehabilitation (CR) has been suggested to improve the preoperative status of the patient and therefore be potentially effective in preventing postoperative complications. The aim of CR is “to influence favourably the underlying cause of cardiovascular disease, as well as to provide the best possible physical, mental and social conditions”[9].’ (p.3, line: 63-66). 

- The added sentence ‘A prehabilitation program consisting of more components (e.g. whole body exercise, dietary, and psychological guidance) can be effective on multiple aspects and might therefore be more effective than a single-component program.’ also contributes to the rationale for conducting the current study (p.3-4, line: 70-72). 

- Also, add the hypothesis of the study.

Reply: Thank you for the suggestion, we added ‘Although this is an exploratory study, based on the shown effects of preoperative IMT[12,13], we expect a lower rate of in-hospital complications and no unintended consequences in patients who followed multidisciplinary prehabilitation.’ to the end of the Introduction (p.3, line: 78-80). 

Methods:

- The methods cannot be followed in this frame.

- Kindly, re-frame the methods in accordance with components (SPICES) for methods:

a. Study design, setting, sample size

b. Participant

c. Intervention/issue of interest (exposure)

d. Comparison

e. Ethics and endpoint

f. Statistical analysis

Reply: Thank you for the suggestion. As suggested, we reframed the section ‘MATERIALS AND METHODS’ in accordance with the SPICES components for methods and we reordered several sentences. Please find a detailed description of the changes to the manuscript below. These changes are also shown in the manuscript with track changes.

Changes to manuscript:

1. We added the header ‘STUDY DESIGN, SETTING, AND SAMPLE SIZE’ (p.4, line: 82).

2. The sample size ‘880’ has been added to the first line of the section ‘MATERIALS AND METHODS’ (p.4, line: 83).

3. We removed ‘(≥18 years)’, ‘(i.e. no main stem stenosis, unstable angina pectoris or progressive symptoms)’, and ‘(i.e. coronary artery bypass graft [CABG], valve surgery, aortic surgery or combined procedures)’ from the first sentence of the section ‘MATERIALS AND METHODS’ (p.4, line: 83-85). The last two removed parts ‘(i.e. no main stem stenosis, unstable angina pectoris or progressive symptoms),’ and ‘(i.e. coronary artery bypass graft [CABG], valve surgery, aortic surgery or combined procedures)’ were included to the first sentence of the new header ‘PARTICIPANT’ and ‘(≥18 years)’ is changed into adults, see next item (item 4).

4. The header ‘PARTICIPANTS’ and the sentence ‘Adults admitted for --- procedures were included.’ were added to the section ‘MATERIALS AND METHODS’ (p.4-5, line: 95-97).

5. The sentences ‘The Heart Center of the UMCG --- the standard hospital protocol of the UMCG.’ were moved from the old header ‘PROCEDURE’ (p.5, line: 109-113) to the new header ‘STUDY DESIGN, SETTING, AND SAMPLE SIZE’ (p.4, line: 86-91). 

6. ‘However, in February 2015 a multidisciplinary preoperative and postoperative rehabilitation program (the Heart-ROCQ pilot program) was implemented in the UMCG for internally referred patients.’ was moved to the header ‘STUDY DESIGN, SETTING, AND SAMPLE SIZE’ (p.4, line: 91-92). 

7. The sentences ‘This resulted in a --- during the waiting time.’ were added at the end of the header ‘STUDY DESIGN, SETTING, AND SAMPLE SIZE’ (p.4, line: 92-94).

8. We changed the header ‘PROCEDURE’ into ‘INTERVENTION’ (p.5, line: 108) and we added the header ‘COMPARISON’ (p.5, line:122).

9. The last sentence of the header ‘INTERVENTION’: ‘Patients who followed this program were referred to as the PRE-group.’ was changed into ‘Patients who followed this program in addition to the standard surgical care were referred to as the PRE-group.’ and was shifted one sentence forwards from line 119,120 to line 117,118 (p.5).

10. The sentences ‘Before February 2015, no prehabilitation was offered --- an outpatient CR was offered as standard care in the referral hospital[14].’ were moved to the new header ‘COMPARISON’ (p.6, line: 123-124). 

11. The words ‘and thus followed only the standard surgical care,’ we added to the last sentence of the new header ‘COMPARISON’ (p.6, line: 126).

12. We changed the header ‘OUTCOMES’ into ‘ETHICS AND ENDPOINT’ (p.6, line: 142).

13. The sentences ‘All collected data were de-identified --- the database of the intensive care unit) and medical records.’ were moved from ‘MATERIALS AND METHODS’ (p.5, line: 103-106) to ‘ETHICS AND ENDPOINT’ (p.7, line: 150-153).

14. We changed the header ‘DATA ANALYSES’ into ‘STATISTICAL ANALYSES’ (p.8, line: 159).

Discussion:

Kindly, reframe it as the following:

a. Main findings of the present study

b. Comparison with other studies

c. Implication, and explanation of findings

d. Strengths and limitations

e. Conclusion, recommendation and future direction.

Reply: Thank you for this suggestion. We carefully went through the Discussion to check the order of topics, with the intention to follow up on this reviewer’s suggested order. However, we came to the conclusion that the current form of the Discussion almost has the same order as suggested. Only within topics ‘b’ and ‘c’ (i.e. ‘comparison with other studies’ and ‘implication, and explanation of findings’) are atrial fibrillation first described (second paragraph, p.13,14, line: 236-245) and followed by the other postoperative complications (third paragraph, p.14, line: 246-260). In addition, we chose to describe the limited sample size as a limitation in the third paragraph (p.14, line: 247-250) and not in the fourth paragraph, to improve the ‘flow’ of the text. 

The order of the current Discussion section is thus (see also the remarks in the revised manuscript with tracking changes):

Paragraph 1:

 a. Main findings of the present study (p.13, line: 231-235)

Paragraph 2:

 b. Comparison with other studies – Atrial Fibrillation (p.13, line: 236-238)

 c. Implication, and explanation of findings – Atrial Fibrillation (p.13, line: 237-238 & p.13,14, line: 239-245)

Paragraph 3:

 b. Comparison with other studies – other postoperative complications (p.14, line: 250-253)

 c. Implication, and explanation of findings – other postoperative complications (p.14, line: 253-260)

Paragraph 4:

 d. Strengths and limitations (p.14-15, line: 261-270)

Paragraph 5:

 e. Conclusion, recommendation and future direction (p.15, line: 274-279).

Other changes to the manuscript:

- The last sentence of the third paragraph within the Discussion section ‘Results on the feasibility and trainability of our prehabilitation program will be submitted soon.’ is changed into ‘Results on the feasibility and trainability of our prehabilitation program are submitted.’ (p.14, line: 260).

7. PLOS authors have the option to publish the peer review history of their article (what does this mean?). If published, this will include your full peer review and any attached files.

Do you want your identity to be public for this peer review? For information about this choice, including consent withdrawal, please see our Privacy Policy.

Reviewer #1: Yes: Salah Altarabsheh

Reviewer #3: Yes: Walid Kamal Abdelbasset

---

## [Decision Letter · Decision Letter 2]

7 Jun 2021

Prehabilitation to prevent complications after cardiac surgery - A retrospective study with propensity score analysis

PONE-D-20-40737R2

Dear Dr. Hartog,

We’re pleased to inform you that your manuscript has been judged scientifically suitable for publication and will be formally accepted for publication once it meets all outstanding technical requirements.

Kind regards,

Academic Editor

PLOS ONE

Additional Editor Comments (optional):

Reviewers' comments:

Reviewer's Responses to Questions

**Comments to the Author**

1. If the authors have adequately addressed your comments raised in a previous round of review and you feel that this manuscript is now acceptable for publication, you may indicate that here to bypass the “Comments to the Author” section, enter your conflict of interest statement in the “Confidential to Editor” section, and submit your "Accept" recommendation.

Reviewer #1: All comments have been addressed

Reviewer #3: All comments have been addressed

2. Is the manuscript technically sound, and do the data support the conclusions?

Reviewer #1: Yes

Reviewer #3: Yes

3. Has the statistical analysis been performed appropriately and rigorously? 

Reviewer #1: Yes

Reviewer #3: Yes

4. Have the authors made all data underlying the findings in their manuscript fully available?

Reviewer #1: Yes

Reviewer #3: Yes

5. Is the manuscript presented in an intelligible fashion and written in standard English?

Reviewer #1: Yes

Reviewer #3: Yes

6. Review Comments to the Author

Reviewer #1: Dear the authors

Thank you for your responses to the reviewers and editorial comments

I have no concerns about this manuscript

Reviewer #3: I would like the authors for taking all considerations for my comments. I have no additional comments

7. PLOS authors have the option to publish the peer review history of their article (what does this mean?). If published, this will include your full peer review and any attached files.

Reviewer #1: **Yes: **Salah Eldien Altarabsheh

Reviewer #3: **Yes: **Walid Kamal Abdelbasset

---

## [Editor Report · Acceptance letter]

7 Jul 2021

PONE-D-20-40737R2 

Prehabilitation to prevent complications after cardiac surgery - A retrospective study with propensity score analysis 

Dear Dr. Hartog:

I'm pleased to inform you that your manuscript has been deemed suitable for publication in PLOS ONE. Congratulations! Your manuscript is now with our production department. 

Kind regards, 

on behalf of

Dr. Robert Jeenchen Chen 

Academic Editor

PLOS ONE